# Effects of Neuromuscular Electrical Stimulation and Therapeutic Ultrasound on Quadriceps Contracture of Immobilized Rats

**DOI:** 10.3390/vetsci11040158

**Published:** 2024-04-01

**Authors:** Kanokwan Suwankanit, Miki Shimizu

**Affiliations:** 1Department of Veterinary Diagnostic Imaging, Faculty of Agriculture, Tokyo University of Agriculture and Technology, 3-5-8 Saiwai-cho, Fuchu 183-0054, Tokyo, Japan; s212485w@st.go.tuat.ac.jp; 2Department of Clinical Sciences and Public Health, Faculty of Veterinary Science, Mahidol University, Nakhon Pathom 73170, Thailand

**Keywords:** therapeutic ultrasound, neuromuscular electrical stimulation, muscle atrophy, muscle fibrosis, sarcomere length

## Abstract

**Simple Summary:**

Quadriceps contracture is a condition that can be observed in young dogs, such as a congenital disability or complication following a hindlimb fracture. Although surgical treatments have been reported, it is difficult to restore normal limb function when the disease is advanced in stage. Rehabilitation early in the course of the disease may effectively improve progress. Using a rat model, we investigated the effectiveness of therapeutic ultrasound and neuromuscular electrical stimulation (NMES) for quadriceps contracture treatment. Quadriceps contracture was induced by immobilizing both hindlimbs for 14 days. Then, we compared the effectiveness of four rehabilitation programs, including spontaneous recovery, therapeutic ultrasound, NMES, and a therapeutic ultrasound and NMES combination on quadriceps contracture for 28 days. After completing 28 days of rehabilitation, the range of joint motion (ROM), muscle histopathology, and levels of fibrosis and anti-fibrosis-associated mRNA expression were measured. The results showed that the therapeutic ultrasound and NMES combination inhibited sarcomere length shortening, muscle atrophy, and muscle fibrosis in quadriceps contracture.

**Abstract:**

Quadriceps contracture is a condition where the muscle–tendon unit is abnormally shortened. The treatment prognosis is guarded to poor depending on the progress of the disease. To improve the prognosis, we investigated the effectiveness of therapeutic ultrasound and NMES in treating quadriceps contracture in an immobilized rat model. Thirty-six Wistar rats were randomized into control, immobilization alone, immobilization and spontaneous recovery, immobilization and therapeutic ultrasound, immobilization and NMES, and immobilization and therapeutic ultrasound and NMES combination groups. The continuous therapeutic ultrasound (frequency, 3 MHz, intensity 1 W/cm^2^) and NMES (TENS mode, frequency 50 Hz; intensity 5.0 ± 0.8 mA) were performed on the quadriceps muscle. On Day 15, immobilization-induced quadriceps contracture resulted in a decreased ROM of the stifle joint, reduction in the sarcomere length, muscle atrophy, and muscle fibrosis. On Day 43, therapeutic ultrasound, NMES, and combining both methods improved muscle atrophy and shortening and decreased collagen type I and III and α-SMA protein. The combination of therapeutic ultrasound and NMES significantly reduced the mRNA expression of *IL-1β*, *TGF-β1*, and *HIF-1α* and increased *TGF-β3*. Therefore, the combination of therapeutic ultrasound and NMES is the most potent rehabilitation program for treating quadriceps contracture.

## 1. Introduction

Quadriceps contracture often occurs in young dogs due to congenital deformities or acquired factors, such as distal femoral fracture, inadequate and prolonged hindlimb immobilization, and tissue trauma during surgery [1,2,3,4,5]. The pathogenesis of quadriceps contracture is an abnormal quadriceps muscle shortening characterized by muscle atrophy and fibrosis and a limited range of stifle joint motion [1,2,6]. Decreased muscle extensibility by muscle immobilization in a shortening position can restrict joint motion, decrease muscle length, and increase collagen expression [7,8,9]. The main pathological changes in muscle contracture in the early stage are decreased muscle fiber size, shortening of muscle fiber, and changes in intramuscular connective tissue [9,10]. Previous studies determined that after joint immobilization, myogenic contracture occurred in the early stage of the muscle contracture (≤2 weeks of immobilization). In contrast, arthrogenic contracture occurred in the late stage of the disease (>2 weeks of immobilization) [11,12,13]. Although there are various surgical techniques for quadriceps contracture treatment [14,15], the prognosis of surgical treatment is guarded to poor depending on the progression of muscle contracture [1,4,5,15,16]. In some cases, amputation is the outcome when the quadriceps contracture progresses to a late stage [3,16]. If initiated early in the course of the disease, rehabilitation for quadriceps contracture can be expected to control disease progression, avoid surgery, and improve prognosis [2,4].

Therapeutic ultrasound is applied for joint contracture in rehabilitation programs [17]. It uses sound waves to utilize the thermal and mechanical effects on the muscle, leading to increased collagen extensibility [18,19,20], improved blood circulation [21,22], and increased range of motion (ROM) [23,24]. Okita et al. (2009) [25] determined that continuous ultrasound waves during joint immobilization can inhibit the deterioration of muscle contracture. However, no other studies show therapeutic ultrasound’s effectiveness alone after immobilization. Results are inconsistent regarding the effectiveness of combining therapeutic ultrasound and stretching. Cunha et al. (2012) [26] found that therapeutic ultrasound with stretching did not address the reduction in muscle weight after immobilization. Usuba et al. (2006) [23] reported that therapeutic ultrasound and stretching inhibited rats’ quadriceps contracture induced by stifle joint immobilization.

Neuromuscular electrical stimulation (NMES) is another helpful modality in rehabilitation treatment for musculoskeletal diseases [17,27]. NMES is one type of transcutaneous electrical nerve stimulation. The innervated nerve stimulates the muscle through surface electrodes at the transcutaneous tissue [17]. It can assist in muscle re-education [28], improve muscle tone [29,30], prevent muscle atrophy [31,32,33], and enhance joint movement [34]. However, the effectiveness of NMES in musculoskeletal tissue is still controversial. Yoshimura et al. (2017) [35] demonstrated that cyclic muscle twitch contraction NMES appears to relieve muscle fibrosis, improve joint motion, and reduce muscle contracture of the sartorius muscle in rats during ankle joint immobilization. However, Williams et al. (1988) [36] showed that NMES can prevent an increase in muscular connective tissue but cannot prevent the loss of sarcomere length resulting from joint immobilization in a shortened position.

Therapeutic ultrasound and NMES relieve joint restriction, muscle atrophy, and muscle fibrosis. Devrimsel et al. (2019) [37] compared the effect of therapeutic ultrasound and NMES on osteoarthritis cases. They found that therapeutic ultrasound can improve muscle stiffness and function, while NMES has more impact on muscle architecture improvement. Amjad et al. (2016) [38] also showed that therapeutic ultrasound was more effective in improving cervical motion range than electrical therapy. The combination of therapeutic ultrasound and NMES has been reported to improve joint function and relieve pain in osteoarthritis patients effectively [39,40]. There is low evidence of therapeutic ultrasound combined with NMES regarding muscle contracture.

The aim of this study is to investigate the effectiveness of therapeutic ultrasound and NMES in treating quadriceps contracture in an immobilized rat model. We applied four rehabilitation programs (no treatment, therapeutic ultrasound alone, NMES alone, and therapeutic ultrasound combined with NMES) to quadriceps contracture induced by stifle joint immobilization. The improvement of quadriceps contracture was evaluated by the ROM of the stifle joint, quadriceps muscle histopathology, and gene expression levels of *interleukin-1β* (*IL-1β*), *transforming growth factor-β1* (*TGF-β1*), *transforming growth factor-β3* (*TGF-β3*), and *hypoxia-inducible transcription factor-1α* (*HIF-1α*). We hypothesized that because therapeutic ultrasound and NMES are effective in improving joint motion, muscle atrophy, and muscle fibrosis, their combined use is the best program for relieving quadriceps contracture.

## 2. Materials and Methods

### 2.1. Animals and Study Design

A total of 36, 13-week-old male specific pathogen-free (SPF), Wistar rats (286.1 ± 10.5 g) were obtained from Japan SLC International Cooperation (Shizuoka, Japan). The animals were housed in individual cages (25 × 40 × 20 cm) inside a room with a 12 h light–dark cycle at an ambient temperature of 24 ± 2 °C. Food and water were available ad libitum. The ethics review committee approved the experimental protocol for animal experimentation at the Tokyo University of Agriculture and Technology (approval number R05-160). Rats were randomly divided into six groups: the control group (Group C, *n* = 6), immobilization-alone group (Group I, *n* = 6), immobilization and spontaneous recovery group (Group S, *n* = 6), immobilization and therapeutic ultrasound group (Group U, *n* = 6), immobilization and NMES treatment group (Group N, *n* = 6), and immobilization and therapeutic ultrasound and NMES combination group (Group UN, *n* = 6).

The timeline of the experiment is shown in Figure 1. The day immobilization started was set as Day 1. All rats were measured for the ROM of the stifle joint before immobilization. The bilateral hindlimbs of rats in Groups I, S, U, N, and UN were then immobilized for 14 days. On Day 15, the bandage was removed, and the ROM of the stifle joint was measured. The rats in Groups C and I were then euthanized, the quadriceps muscles were harvested, and the quadriceps muscles were evaluated histopathologically for quadriceps contractures. In Groups U, N, and UN, a 28-day rehabilitation treatment started. Rats in Group S did not receive treatment. On Day 43, rats in Groups S, U, N, and UN were measured for the ROM of the stifle joint and the creatinine phosphokinase level and euthanized. After euthanasia, quadriceps muscles were harvested.

### 2.2. Immobilization Procedure

All rats were anesthetized with isoflurane using an open-drop chamber technique. Anesthesia was maintained in an anesthetic machine with isoflurane at an oxygen flow rate of 300–600 mL/min. During anesthesia, the rats’ muscles were completely relaxed and unresponsive to stimulation. The rats’ respiratory rate was maintained at a stable rate (50–100 bpm). Rats’ fur at both hindlimbs was removed. Following the wire immobilization method [10], the rats in Groups I, S, U, N, and UN were immobilized by applying bilateral bandages at the stifle joint full extension and ankle plantar flexion positions. A steel bonsai wire (diameter 2.5 mm, length 70 cm) was set around the fourth and fifth lumbar vertebrae and wrapped around both hindlimbs from the hip joint to the toe. Non-elastic bandage tape (Multipore^TM^ Sports White Athletic Tape, 3M Japan or Kinesio^®^ tape, Tokyo, Japan) was also applied before the wire application to prevent dermatitis. Rats in Group C were not immobilized (from Day 1 to Day 15).

All rats were fitted with a modified vest collar [41]. And all rats were capable of daily activities. Any clinical signs of bandage loosening or adverse events such as hindlimb edema and necrosis were observed once a day. They were changed if the bandages became loose or the rats showed any signs of hindlimb injury. At the end of the 14-day immobilization period (Day 15), the bandages were removed. Rehabilitation treatment programs were then started in Groups U, N, and UN.

### 2.3. Spontaneous Recovery

Rats in Group S did not receive treatment between Day 15 and Day 42 after the bandage removal. However, to match conditions with the treatment group, they were anesthetized for 15 min once a day, 5 days a week, for 28 days.

### 2.4. Therapeutic Ultrasound

Therapeutic ultrasound treatment was conducted for 28 days, from Day 15 to Day 42. In Groups U and UN, treatment was performed with a commercially available multi-channel electrotherapy and ultrasound combo device (ESTIMUS; Ito Physiotherapy and Rehabilitation, Tokyo, Japan, Figure 2).

Rats in Groups U and UN underwent general anesthesia with isoflurane by using an induction chamber and maintenance with a face mask and anesthesia machine. Rats were placed in lateral recumbency with the leg of the treatment side uppermost. The bilateral quadriceps muscles were treated with therapeutic ultrasound (continuous wave; intensity 1 W/cm^2^, frequency 3 MHz). Our pilot study confirmed that during ultrasound treatment, the temperature of the quadriceps muscle increased at a frequency of 3 MHz and an intensity of 1 W/cm^2^ by 2–4 °C from the baseline muscle temperature. The transducer head diameter was 16 mm, and the effective radiating area was 0.7 cm^2^. Aqueous gel was utilized as a coupling medium. The ultrasound transducer head was slowly moved circularly over the entire quadriceps muscle to deliver ultrasonic irradiation. The quadriceps muscle was irradiated for 30 min on each side once a day, 5 days a week, for 28 days. Therapeutic ultrasound was performed on rats in Group UN before performing NMES.

### 2.5. Neuromuscular Electrical Stimulation (NMES)

Electrotherapy was conducted for 28 days, from Day 15 to Day 42. In Groups N and UN, treatment was performed using a commercially available multi-channel electrotherapy and ultrasound combo device (ESTIMUS, Figure 2). During performing the NMES, rats were anesthetized with isoflurane. The bilateral quadriceps muscles were subjected. The inguinal region and 1 cm of the suprapatellar border were shaved, and four self-adhesive electrodes (circular, 1 cm in diameter) were attached to both hindlimbs together. The rats were placed in a supine position, and hindlimbs were extended. The stifle and hip joint were maintained in a natural position. NMES (TENS mode; surge co-continuous; alternating current; frequency 50 Hz; phase duration 100 µs; time on 5 s; time off 20 s; ramp time 1 s [31,33,42]) was applied for 15 min once a day, 5 days per week, for 28 days. The intensity was set at 5.0 ± 0.8 mA, depending on the degree of muscle contraction.

### 2.6. ROM of the Stifle Joint

Before immobilization, after immobilization (Day 15), and after rehabilitation (Day 43), the rats were measured for the degree of flexion and extension of the stifle joint by using a goniometer following Millis et al. (2014) [43]. The ROM of the stifle joint was calculated by deducting the angle of the flexed position from that of the extended position.

### 2.7. Body Weight and Food Intake

Body weight and food intake were measured in Groups S, U, N, and UN to assess stress in rats during the rehabilitation period (from Day 15 to Day 43). Body weight was measured at 08.00 am on the day of treatment. Food intake was measured daily from Day 15 to Day 43.

### 2.8. Evaluation of Creatinine Phosphokinase Level

Creatinine phosphokinase levels were measured at Day 43 in Groups S, U, N, and UN to detect muscle damage due to the rehabilitation programs. The blood of rats was collected from their heart by the cardiac puncture method while they were alive under isoflurane anesthesia. A heparin tube was used to collect blood and then centrifuged to obtain a serum. Then, a DRI-CHEM NX700 analyzer (Fujifilm Medical Co., Ltd., Tokyo, Japan) was used to measure creatinine phosphokinase levels.

### 2.9. Euthanasia, Muscle Collection, and Measurement

Rats in Groups C and I were euthanized after completing 14 days of the immobilization period (on Day 15). Rats in Groups S, U, N, and UN were euthanized on Day 43. An open-drop technique with 30% isoflurane was applied for euthanasia. According to the Guide for the Care and Use of Laboratory Animals [44], cardiac arrest confirmation was performed by using bilateral thoracotomy.

Bilateral quadriceps muscles were removed, and a digital scale was used to measure wet weight. For evaluation of muscle mass, body weight was used to standardize the quadriceps muscle’s wet weight. Quadriceps muscle size (length × width × height) was measured using a digital caliper.

### 2.10. Histopathological Examination

The right quadriceps muscle was cut into the longitudinal section and cross-section and then fixed in 4% paraformaldehyde at 4 °C for 36–48 h. The specimens were rinsed in distilled water, dehydrated in a graded series of ethanol, equilibrated with xylene, and embedded in paraffin. Serial 4 µm cross-sections and longitudinal sections were cut with a microtome and stained with hematoxylin-eosin (H&E) [45].

The histological sections were observed under a light microscope (Keyence BZ-X800, Keyence Corporation, Osaka, Japan) at a final magnification of ×400 to measure muscle fiber size and sarcomere length. Morphological analyses were analyzed using ImageJ software version 1.53 (National Institutes of Health, Bethesda, MD, USA). By random selection, the muscle fiber’s cross-sectional area and minimum Feret’s diameter were measured from 400 muscle fibers. Over 1000 sarcomeres were randomly selected from 10 longitudinal images per quadriceps muscle for sarcomere length measurement.

The histopathological grading score of muscle damage was evaluated using the information on four domains: inflammation, fibrosis, vasculitis, and muscle degeneration, based on the 2007 international consensus proposed score system for muscle biopsy [46,47]. A total area of muscle tissue sample was carefully observed under a light microscope (Axiostar plus, Carl Zeiss, Goettingen, Germany) at a magnification of ×200 by an experienced pathologist, blinded to the experimental group to which the slides belonged. Table 1 shows each domain’s muscle pathological biopsy grading score scale.

### 2.11. Immunohistochemical Analysis

Tissue sections underwent deparaffinization in xylene and subsequent rehydration through a series of different ethanol grades. The tissue slides were placed in an autoclave for a target retrieval solution at 121 °C for 10 min and then incubated with 1% H_2_O_2_ in methanol for 30 min at room temperature. After being washed in 0.01 M phosphate-buffered saline (PBS; pH 7.4), the sections were blocked with 1% normal goat serum for 30 min at room temperature. The sections were then incubated overnight at 4 °C with collagen type I monoclonal antibody (COL-1; Invitrogen^®^, Carlsbad, CA, USA; diluted ×200), collagen type III monoclonal antibody (FH-7A, Invitrogen^®^, Rockford, IL, USA; diluted ×50), and monoclonal mouse anti-human smooth muscle actin (Clone 1A4, Sigma-Aldrich^®^, Glostrup, Denmark; diluted ×200). The sections were rinsed in PBS, after which Envision horseradish peroxidase anti-mouse (Dako EnVision+, HRP, Agilent Technology^®^, Santa Clara, CA, USA) was applied at room temperature for 30 min. After being washed in PBS, horseradish peroxidase-binding sites were visualized with 0.05% 3,3′-diaminobenzidine and 0.01% H_2_O_2_ in 0.05 M Tris buffer at room temperature. The sections were rinsed in PBS for the final washing step before being counterstained with hematoxylin. Then, they were dehydrated in increasing alcohol concentrations, washed with xylene, and mounted on slides. A light microscope (Keyence BZ-X800, Keyence Corporation, Osaka, Japan) and ImageJ software version 1.53 (National Institutes of Health, Bethesda, MD, USA) were utilized for immunohistochemical analysis. The myofibroblast number was determined by measuring the α-smooth muscle actin (α-SMA)-positive cells. The number of α-SMA-positive cells was counted in all fields for each section at ×200 magnification, and the average number per field was calculated. The percentage of collagen types I and III in the total tissue longitudinal area, which comprises the muscle, perivascular, and interstitial regions, was analyzed at ×200 magnification. Areas of muscle overlap or poor staining were excluded. All analyses were blinded to the experimental group to which the slides belonged.

### 2.12. Real-Time RT-PCR to Measure Interleukin-1β (IL-1β), Transforming Growth factor-β1 (TGF-β1), Transforming Growth Factor-β3 (TGF-β3), and Hypoxia-Inducible Transcription Factor-1α (HIF-1α) mRNA

We assessed the improvement of muscle fibrosis after rehabilitation by detecting mRNA expression of *IL-1β*, *TGF-β1*, *TGF-β3*, and *HIF-1α* using real-time RT-PCR in the left quadriceps muscle of Groups S, U, N, and UN. TRIzol reagent (TRIzol^TM^, Invitrogen^®^, Carlsbad, CA, USA) was applied to extract total RNA from the muscle samples, according to the manufacturer’s protocol. For cDNA preparation, total RNA was used as a template with qPCR RT Master Mix (ReverTra Ace, Toyobo Co., Ltd., Osaka, Japan). After that, the mRNA expression levels were determined by a real-time RT-PCR performed in an optical 96-well plate with the StepOnePlus^TM^ Real-Time PCR System (Applied BioSystems, Foster City, CA, USA). The sequences of primers used for the real-time RT-PCR are shown in Table 2. Relative quantification was calculated using the 2^−∆∆Ct^ method [33] and normalized to *β-actin*. Data are presented as expression levels relative to the expression level in Group S.

### 2.13. Statistical Analysis

Statistical analyses were performed using GraphPad Prism Version 8 (GraphPad Software Inc., San Diego, CA, USA). Mean ± standard deviation (SD) is used to present parametric data. The differences between groups were performed using a one-way analysis of variance (ANOVA), followed by Tukey’s multiple comparisons test. Non-parametric data are presented as medians and interquartile ranges. The differences between the two time points within the same group were assessed using the Wilcoxon test, and differences between three time points were assessed using the Friedman test and Dunn’s multiple comparison tests. Differences between groups were analyzed using the Kruskal–Wallis and Dunn’s multiple comparison tests. The differences were considered significant at *p* < 0.05.

## 3. Results

### 3.1. Stifle Joint Angle and ROM

Figure 3 shows the stifle joint ROM measurements and angles at maximum extension and flexion before immobilization, after the immobilization period (Day 15), and after the rehabilitation period (Day 43).

Comparisons of 2 time points between before immobilization and Day 15 within Groups C and I were assessed using the Wilcoxon test, and comparisons of 3 time points within Groups S, U, N, and UN were assessed using the Friedman test and Dunn’s multiple comparison tests. The extension angle on Day 15 was lower than before immobilization in Groups I, S, and U (*p* = 0.0059, 0.0033, and 0.0128, respectively), while it was unchanged in Groups N and UN. The flexion angles on Day 15 were higher than before immobilization in Groups I, S, U, N, and UN (*p* = 0.0001, 0.0016, 0.0016, 0.0023, and 0.0047, respectively). As a result, the ROM on Day 15 was lower than before immobilization in Groups I, S, U, N, and UN (*p* = 0.0001, 0.0023, 0.0128, 0.0092, and 0.0176, respectively). Within-group comparisons were assessed using the Kruskal–Wallis and Dunn’s multiple comparison test. On Day 15, the flexion angle increased in Groups I, S, U, N, and UN compared to Group C (*p* = 0.0007, 0.0005, 0.0001, 0.0043, and 0.0002, respectively), and the ROM decreased (*p* = 0.0005, 0.0002, 0.0001, 0.0100, and 0.0002, respectively).

At Day 43 in Groups S, U, N, and UN, the extension angle was higher (*p* = 0.0016, 0.0011, 0.0092, and 0.0128, respectively), the flexion angles were lower (*p* = 0.0003, 0.0003, 0.0002, and 0.0001, respectively), and the ROM increased (*p* = 0.0002, 0.0001, 0.0001, and 0.0001, respectively) compared to Day 15. Within-group comparison at Day 43 was performed by using the Kruskal–Wallis and Dunn’s multiple comparison test; Group UN had a decreased flexion angle and increased ROM compared to Group S (*p* = 0.0044 and *p* = 0.0001, respectively).

### 3.2. Body Weight and Food Intake

The body weight and food intake of Groups S, U, N, and UN in the rehabilitation period (from Day 15 to Day 43) are represented in Table 3. The comparisons of body weight and food intake within the same group at each time point were assessed using the Friedman test and Dunn’s multiple test, and the comparisons between groups at the same time point were performed by using the Kruskal–Wallis and Dunn’s multiple comparison test. The body weight increased on Day 43 compared to Day 22 in Groups S, U, N, and UN (*p* = 0.0001, 0.0006, 0.0001, and 0.0001, respectively). The food intake did not change during the rehabilitation period.

### 3.3. Creatinine Phosphokinase Level

The creatinine phosphokinase levels of Groups S, U, N, and UN on Day 43 are represented in Table 4. There were no differences in creatinine phosphokinase levels assessed by using the Kruskal–Wallis and Dunn’s multiple comparison test.

### 3.4. The Ratio of Quadriceps Muscle Weight to Body Weight and Quadriceps Muscle Measurements

The ratio of quadriceps muscle weight to body weight and the quadriceps muscle length, width, and height are shown in Table 5. The comparisons of muscle weight to body weight and the quadriceps muscle length, width, and height between groups were assessed using the Kruskal–Wallis and Dunn’s multiple comparison test. Muscle weight to body weight was higher in Groups S, U, N, and UN than in Group I (*p* = 0.0050, *p* = 0.0013, *p* = 0.0001, and *p* = 0.0001, respectively). Groups N and UN were higher than Group C (*p* = 0.0024 and *p* = 0.0001, respectively). Muscle length of Groups I, S, and U was lower than Group C (*p* = 0.0001, *p* = 0.0001, and *p* = 0.0035, respectively), and Groups N and UN were higher than Group I (*p* = 0.0001 in both groups) and Group S (*p* = 0.0253 and *p* = 0.0002, respectively).

### 3.5. Histological Evaluation of Quadriceps Muscle Fiber and Sarcomere Length

Histological findings of the quadriceps muscle in cross-sections with H&E staining in Group C showed a typical microanatomical skeletal muscle tissue arrangement. In contrast, Group I showed a mild to moderate accumulation of inflammatory cells, most of which were lymphocytes and macrophages. There were fibroblastic responses, accumulation of fibrous tissues in the endomysium and perimysium, and a moderate degree of muscle degeneration. In Group S, there was an accumulation of inflammatory cells, fibroblastic responses, and an accumulation of fibrous tissues in the endomysium and perimysium. However, in Groups U, N, and UN, a minimal degree of accumulation of inflammatory cells, minimal fibroblastic responses, and accumulation of fibrous tissues were found.

Histopathological scores, including inflammation, vasculitis, fibrosis, and degeneration of the quadriceps muscle, are provided in Table 6. The comparisons of histopathological scores between groups were assessed using the Kruskal–Wallis and Dunn’s multiple comparison test. Group I scored higher in all scores than Group C. Groups U, N, and UN improved their scores compared to Group I.

The cross-sectional area, minimum Feret’s diameter, and sarcomere length of the quadriceps muscles are shown in Figure 4A. The comparisons of cross-sectional area, minimum Feret’s diameter, and sarcomere length between groups were assessed using the Kruskal–Wallis and Dunn’s multiple comparison test. These parameters were lower in the groups with joint immobilization (Groups I, S, U, N, and UN) than in Group C. And these parameters increased in Groups S, U, N, and UN compared to Group I. Group UN showed the most improvement in muscle atrophy and lengthening in sarcomere among the groups with rehabilitation.

### 3.6. Measurement of α-Smooth Muscle Actin (α-SMA) Cells and Collagen Types I and III

The percentages of collagen types I and III and the number of α-SMA-positive cells are represented in Figure 4B. The comparisons of collagen types I and III percentages and the number of α-SMA-positive cells between groups were assessed using one-way ANOVA and Tukey’s multiple comparisons test. These examination items were higher in Groups I, S, U, N, and UN than in Group C. However, they were lower in Groups S, U, N, and UN than in Group I. Group UN improved these increases the most.

### 3.7. Relative Expression of IL-1β, TGF-β1, TGF-β3, and HIF-1α mRNA

The expression levels of *IL-1*β, *TGF-*β*1*, *TGF-*β*3*, and *HIF-1*α mRNA in quadriceps muscle are represented in Figure 5. The comparisons of *IL-1*β, *TGF-*β*1*, *TGF-*β*3*, and *HIF-1*α mRNA expression levels between groups were assessed using one-way ANOVA and Tukey’s multiple comparisons test. After rehabilitation (Day 43), *IL-1*β mRNA expression in Groups N and UN was lower than in Group S (*p* = 0.0026 and *p* = 0.0028, respectively, Figure 5A). *TGF-*β*1* mRNA expression in Groups N and UN was lower than in Group S (*p* = 0.0153 and *p* = 0.0010, respectively), and Group UN was also lower than Group U (*p* = 0.0259, Figure 5B). *TGF-*β*3* mRNA expression in Group UN was higher than in Group S (*p* = 0.023, Figure 5C). *HIF-1*α mRNA expression in Group UN was lower than in Group S (*p* = 0.0441, Figure 5D).

## 4. Discussion

We evaluated the effectiveness of therapeutic ultrasound and NMES for quadriceps contracture treatment based on the pathogenesis of quadriceps contracture. After immobilization, quadriceps contracture was detected and characterized by the limitation of the ROM of the stifle joint, muscle atrophy, and fibrosis. After rehabilitation, therapeutic ultrasound combined with NMES improved the ROM limitation of the stifle joint, muscle atrophy, and muscle fibrosis rather than no treatment or using them alone.

As in previous reports [10], in this study, 14 days of immobilization induced quadriceps contracture, resulting in a decreased ROM of the stifle joint and an increased flexion angle compared to the control group. A reduction in the ROM would result from myogenic changes [11]. The histological changes in quadriceps contracture are muscle fiber atrophy and a reduction in the sarcomere length because the quadriceps muscle was immobilized in a shortening position [48]. This study found decreased quadriceps muscle fiber size, indicated by cross-sectional area and minimum Feret’s diameter and shortening of sarcomere length in immobilization-alone groups.

Another pathological change in quadriceps contracture is muscle fibrosis, characterized by overexpression of collagen in the endomysium and perimysium of the muscle [6]. The dominant collagen types in intramuscular connective tissue and associated with muscle contracture are collagen types I and III [49]. In this study, the percentages of collagen types I and III in quadriceps muscle were increased in the immobilization-alone groups. *TGF-*β is a cytokine involved in the fibrotic pathway [50]. It includes three isoforms, including *TGF-*β*1*, *2*, and *3* [50,51]. Zhao et al. (2010) [51] found that *TGF-*β*1* has a profibrotic function and *TGF-*β*3* has an anti-fibrotic function in biopsy specimens from gluteal contracture muscle cases. *TGF-*β*1* is a crucial activator of collagen production from fibroblasts, induced by *IL-1*β, and plays an essential role in muscle fibrosis in muscle contracture [52]. Early study shows the upregulation of *TGF-*β*1* and collagen types I and III in gluteal muscle contracture [51]. *TGF-*β*3* has been reported to inhibit fibrosis responses and has an anti-scarring effect in wound healing [50,53]. In human myocardial fibrosis, *TGF-*β*3* is reported to reduce the proliferation and migration of human cardiac fibroblasts, decrease collagen synthesis, and promote myocardial remodeling [54]. In hepatic fibrosis, *TGF-*β*3* has been shown to attenuate the degree of hepatic fibrosis and collagen accumulation [55]. *HIF-1*α is another cytokine related to muscle contracture in a hypoxic response to induce muscle fibrosis [56]. Hypoxia, stimulated by reduced blood perfusion inside the muscle, can induce collagen production by converting the fibroblast to a myofibroblast [57,58]. After immobilization to induce muscle contracture for more than 4 weeks, *HIF-1*α was increased, and collagen accumulation was found in muscle tissue [59]. In muscle contractures, myofibroblasts produce collagen [60]. α-SMA is a marker for myofibroblasts. In the present study, joint immobilization increased the ratio of α-SMA-positive cells to myofibers and collagen types I and III.

Therapeutic ultrasound with a continuous mode has thermal and mechanical effects to increase muscle temperature and circulation and stimulate fracture repair [61,62]. The thermal effect on muscle tissue changes muscle extensibility, improving the ROM restriction [24,61]. Watanabe et al. (2017) [63] reported that therapeutic ultrasound (frequency 3 MHz; intensity 30 mW/cm^2^, 5 days per week for 4 weeks) for stifle joint contracture after 8 weeks of immobilization in rats significantly improved the ROM of the stifle joint. A previous study reported increased muscle extensibility changes resulting from a 3 to 4 °C rise in muscle tissue temperature in dogs [64]. Also, a crucial therapeutic ultrasound variable that impacts muscle temperature is frequency. The frequency ranges from 1 to 3 MHz depending on the depth of the target tissue. A frequency of 3 MHz is used for superficial tissue treatment (depth between 0.5 and 3 cm), while 1 MHz is used for deep tissue treatment (depth between 2 and 5 cm) [61]. Levine et al. (2001) [64] reported that therapeutic ultrasound (frequency 3.3 MHz; intensity 1 W/cm^2^) on caudal thigh dogs’ muscles increased superficial muscle temperature. In our pilot study, we examined a therapeutic ultrasound setting to increase quadriceps muscle temperature by 2–4 °C and decided on a frequency of 3 MHz and an intensity of 1 W/cm^2^.

Therapeutic ultrasound treatment to improve muscle atrophy is still controversial. Sakamoto et al. (2012) [65] reported that continuous therapeutic ultrasound (frequency 1 MHz; intensity 1 W/cm^2^) application to soleus muscle over 2 weeks of immobilization period inhibits a decrease in the muscle fiber diameter via thermal effect. However, Cunha et al. (2012) [26] reported that therapeutic ultrasound (frequency 1 MHz) in different intensities (1.0, 0.5, and 0.2 W/cm^2^) combined with stretching to treat the soleus muscle after immobilization cannot improve muscle weight. Similar to our study, the muscle weight and fiber size showed no significant difference after using therapeutic ultrasound compared to no treatment. In contrast, we found an increase in sarcomere length after using therapeutic ultrasound. Previous studies supported that therapeutic ultrasound has the effect of increasing the hamstring muscle length in healthy humans [66] and increasing the sarcomere length in rats after muscle injury [67]. Additionally, therapeutic ultrasound has anti-fibrotic effects in reducing renal fibrosis on experimental hypertensive nephropathy and diabetic nephropathy in mice [68] and in reducing the density of collagen type I, HIF-1α, and α-SMA after prolonged hypoxia-induced cardiac fibrosis in mice [69]. Our study found that after using therapeutic ultrasound, collagen types I and III percentages and the number of α-SMA-positive cells decreased compared to no treatment.

NMES physically increases muscle strength and tone, reduces muscle spasms, and increases muscle extensibility and ROM [70,71]. NMES can stimulate quadriceps muscle contraction, increasing joint movement [72]. It has been used to protect the reduction in ankle joint motion in rats resulting from soleus contracture induced by immobilization [35]. Honda et al. (2021) [42] reported an improvement in the limitation of the ROM of rats’ ankles after using belt electrode-skeletal muscle electrical stimulation (frequency 50 Hz, 6 days per week) during the 2-week immobilization period. NMES effectively prevents muscle atrophy during immobilization [35] and increases sarcomere length [73]. In this study, using NMES to treat quadriceps contracture improved sarcomere length and muscle fiber size reduction rather than no treatment and therapeutic ultrasound. Additionally, NMES can inhibit muscle fibrosis by decreasing the accumulation of collagen types I and III and reducing *TGF-*β*1* and α-SMA during immobilization-induced soleus contracture in rats [35]. Our result regarding the fibrosis pathway showed that after using NMES, the mRNA expression levels of *IL-1*β and *TGF-*β*1*, the percentages of collagen types I and III, and the number of α-SMA-positive cells decreased compared to no treatment.

Few studies have been reported on the impact of therapeutic ultrasound combined with NMES on muscle tissue. The therapeutic ultrasound and NMES combination increases muscle strength, increases ROM, and improves the daily activities of humans with rotator cuff syndrome [74]. Venosa et al. (2019) [75] reported that combining therapeutic ultrasound and NMES can improve cervical ROM in humans. In this study, the combination of therapeutic ultrasound and NMES (Group UN) improved the ROM of the stifle joint, quadriceps muscle weight, quadriceps muscle length, sarcomere length, and muscle fiber size compared to their use alone. The synergistic effect of therapeutic ultrasound and NMES is presumed to have increased muscle size and extensibility and improved the ROM of the stifle joint. Furthermore, Group UN showed increased expression of *TGF-*β*3* mRNA and decreased expression of *IL-1*β, *TGF-*β*1*, and *HIF-1*α compared to no treatment (Group S). In addition, the percentages of collagen type I and III and the number of α-SMA-positive cells decreased in Group UN compared to when using them alone. This suggested that therapeutic ultrasound combined with NMES may be able to inhibit muscle fibrosis.

Stress can occur after repeated anesthesia in the long term, so all rats must be monitored for signs of stress and illness [76]. Body weight and food intake can guarantee that rats have a normal physiological health status [77]. Although rehabilitation was performed under anesthesia, food intake was in the average dietary intake (15 g/rat/day) [78] in all groups, and there were no significant differences in the body weight and food intake between groups. In clinical settings of dogs and cats, companion animals are familiar to humans and can relax in a comfortable environment [79]. Therapeutic ultrasound and NMES in appropriate settings are painless and do not require sedation or anesthesia [61,70]. However, if the animal is stressed or treatment is limited, stress should be minimized, such as by administering anxiolytics [80].

The important side effect of therapeutic ultrasound and NMES to consider is muscle damage due to overuse and inappropriate settings [61,62,70]. Muscle damage, including muscle inflammation, vasculitis, fibrosis, and degeneration, can be identified using the histopathological scoring system of muscle biopsy, providing semiquantitative data. In this study, the total muscle damage score using therapeutic ultrasound or NMES to treat quadriceps contracture for 28 days was not significantly different from that of the control group, and it was lower than that of immobilization alone. It means that three rehabilitation programs can be used to decrease the incidence of muscle inflammation, vasculitis, fibrosis, and degeneration in quadriceps contracture. In addition to using a histopathological scoring system, the level of serum creatinine phosphokinase can be used to identify muscle damage [81,82]. In this study, the creatinine phosphokinase level in four rehabilitation groups showed no significant difference, meaning the rehabilitation programs did not affect muscle damage.

Our study has several limitations. First, this study did not measure the effects of rehabilitation treatment every week. This is because previous information showed that muscle recovery from muscle contracture takes 2 to 4 times longer than the immobilization period [4,83]. Second, muscle temperature was not assessed during therapeutic ultrasound. This was because of the possibility of traumatizing the muscle by inserting a thermometer needle. We did confirm an increase in muscle temperature in our pilot study. Third, only male rats were used in this study. This is because the effect of sex hormones such as estrogen and progesterone during the estrus cycle may affect muscle maintenance and growth [84]. Finally, in the present study, the ROM of the stifle joint improved without treatment. It is presumed that the limitation of ROM of the stifle joint due to the 2-week immobilization period would improve with daily activities after the immobilization was released. Future studies will evaluate the effects of ultrasound therapy and NMES in conditions where the ROM does not improve with no treatment and will assess the thermal and mechanical effects of therapeutic ultrasound and the muscle contraction effect of NMES on quadriceps contracture.

## 5. Conclusions

These results suggested that 14 days of immobilization of quadriceps muscle-induced quadriceps contracture resulted in a decrease in the ROM of the stifle joint, reduction in the sarcomere length, muscle atrophy, and muscle fibrosis. It was suggested that the combination of therapeutic ultrasound and NMES promotes recovery from quadriceps contracture. This is due to improving the limitation in the stifle joint motion, increasing the muscle fiber size and sarcomere length, and decreasing type I and III collagen accumulation. Also, they can decrease *IL-1β* and *TGF-*β*1* mRNA expression levels, which are associated with muscle fibrosis, and increase the expression levels of *TGF-*β*3* mRNA related to anti-fibrosis. Combination therapy of therapeutic ultrasound and NMES is superior to their use alone. These rehabilitation methods can be included and used in a comprehensive method of treating the early stage of quadriceps contracture.

## Figures and Tables

**Figure 1 vetsci-11-00158-f001:**
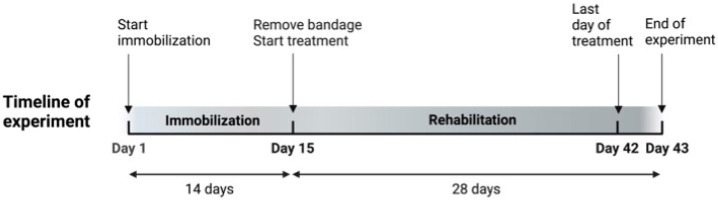
Timeline of the experiment.

**Figure 2 vetsci-11-00158-f002:**
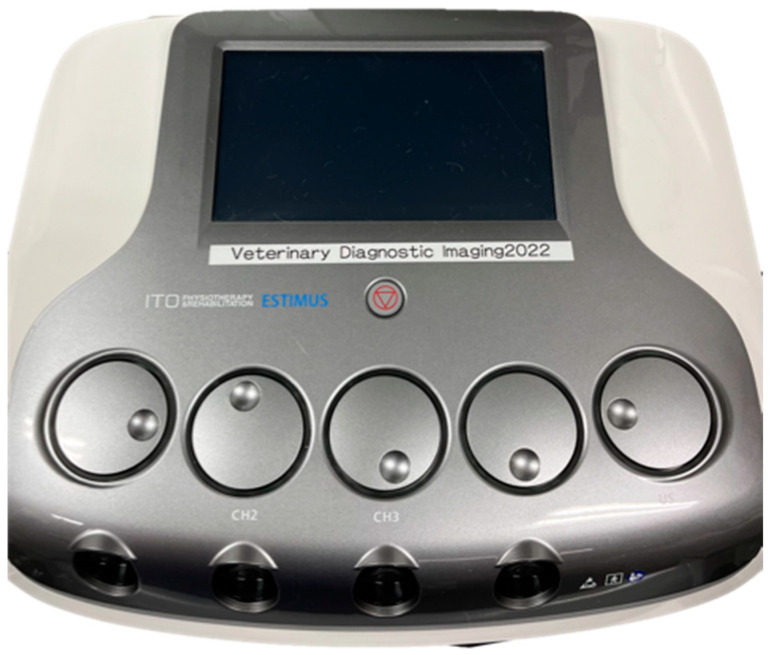
An image of ESTIMUS.

**Figure 3 vetsci-11-00158-f003:**
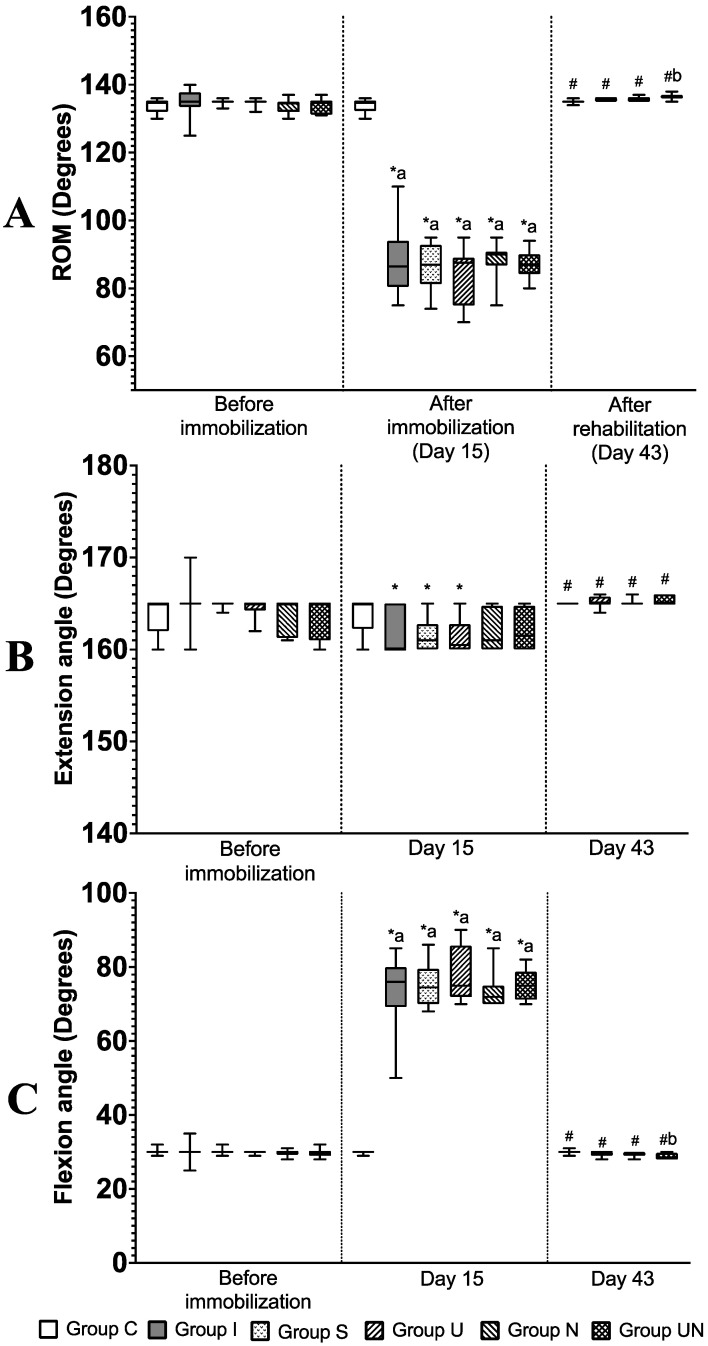
(**A**) ROM of the stifle joint (°), (**B**) extension angle, and (**C**) flexion angle of the stifle joint of rats. Values are expressed as median with interquartile ranges. Group C; control group (*n* = 12); Group I, immobilization-alone group (*n* = 12); Group S, immobilization and spontaneous recovery group (*n* = 12); Group U, immobilization and therapeutic ultrasound group (*n* = 12); Group N, immobilization and NMES group (*n* = 12); Group UN, immobilization and therapeutic ultrasound and NMES combination group (*n* = 12). *, significant difference from before immobilization in the same group (*p* < 0.05); ^#^, significant difference from Day 15 in the same group (*p* < 0.05); a, significant difference from Group C in the same period (*p* < 0.05); b, significant difference from Group S in the same period (*p* < 0.05).

**Figure 4 vetsci-11-00158-f004:**
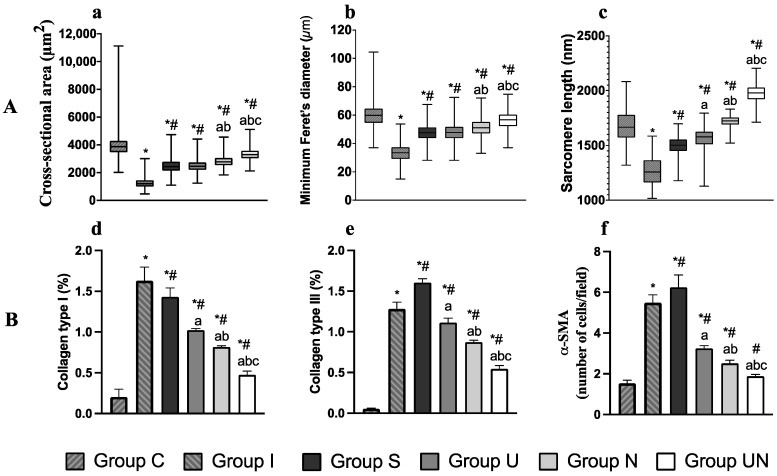
(**A**) Histological analysis of quadriceps muscles using H&E staining. Values are expressed as medians with interquartile ranges. (**a**) cross-sectional area, (**b**) minimum Feret’s diameter, and (**c**) sarcomere length. (**B**) Immunohistochemical analysis of quadriceps muscle. Values are expressed as mean ± standard deviation. (**d**) the percentage of collagen type I, (**e**) the percentage of collagen type III, and (**f**) the number of α-smooth muscle actin (α-SMA)-positive cells. Group C, control group; Group I, immobilization-alone group; Group S, immobilization and spontaneous recovery group; Group U, immobilization and therapeutic ultrasound group; Group N, immobilization and NMES group; Group UN, immobilization and therapeutic ultrasound and NMES combination group; *, significant difference from Group C (*p* < 0.05); ^#^, significant difference from Group I (*p* < 0.05); a, significant difference from Group S (*p* < 0.05); b, significant difference from Group U (*p* < 0.05), c, significant difference from Group N (*p* < 0.05).

**Figure 5 vetsci-11-00158-f005:**
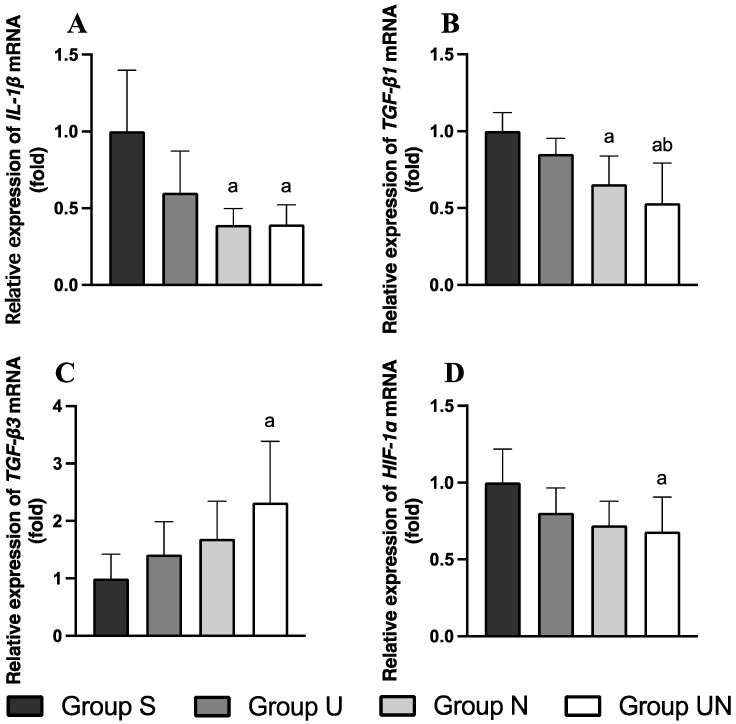
Expression levels of *IL-1β* (**A**), *TGF-β1* (**B**), *TGF-β3* (**C**), and *HIF-1α* (**D**) mRNA. Values are expressed as mean ± standard deviation. Group S, immobilization and spontaneous recovery group (*n* = 6); Group U, immobilization and therapeutic ultrasound group (*n* = 6); Group N, immobilization and NMES group (*n* = 6); Group UN, immobilization and therapeutic ultrasound and NMES combination group (*n* = 6); a, significantly different from Group S (*p* < 0.05); b, significantly different from Group U (*p* < 0.05).

**Table 1 vetsci-11-00158-t001:** Muscle biopsy scores.

Histopathological Parameters	Score			
	0	1	2	3
Inflammation Inflammatory cell infiltration (leukocyte accumulation)	None (<10 cells at 20× high power field)	1 cluster (≥10 cells at 20× high power field)	2 clusters	≥3 clusters
Vasculitis	None	1 vessel	2 vessels	3 vessels
Fibrosis Collagen is present as pale eosinophilic fibrillar material in perimysium and endomysium	Normal level	Mild increase	Moderate increased deposition separating and surrounding adjacent fibers with attenuation of several muscle fibers	Marked presence of collagen deposits in large areas and loss of tissue architecture
Muscle fiber degeneration Degeneration is characterized by irregular shapes, cell swelling, or cell shrunken, hypereosinophilia (pale color), and necrosis	<5% of total area	6–25%	26–50%	>50%

**Table 2 vetsci-11-00158-t002:** Sequences of primers used for the real-time RT-PCR.

Object Gene	F/R	Sequence
*IL-1* β	F	5′-AATGACCTGTTCTTTGAGGCTGAC-3′
	R	5′-CGAGATGCTGCTGTGAGATTTGAA-3′
*TGF-* β *1*	F	5′-ATGCCAACTTCTGTCTGGGG-3′
	R	5′-GGTTGTAGAGGGCAAGGACC-3′
*TGF-* β *3*	F	5′-GGACTTCGGCCACATC-3′
	R	5′-CGGGTGCTGTTGTAAA-3′
*HIF-1* α	F	5′-TGCTGGCTCCCTATATCCCA-3′
	R	5′-GGAGGGCTTGGAGAATTGCT-3′
*β-actin*	F	5′-GCAGGAGTACGATGAGTCCG-3′
	R	5′-ACGCAGCTCAGTAACAGTCC-3′

F, forward; R, reverse.

**Table 3 vetsci-11-00158-t003:** Body weight (g) and food intake (g) of rats from Day 15 to Day 43 in the rehabilitation period.

Parameters	Day	Group			
		S	U	N	UN
Body weight	15	250.3 (233.4–258.4)	251.3 (241.5–262.8)	256.5 (235.0–263.4)	252.3 (247.0–266.6)
	22	231.0 (225.0–236.8)	230.3 ^a^ (213.0–239.8)	232.5 (223.5–235.1)	237.8 (232.0–244.0)
	29	246.0 (238.1–255.3)	233.5 (227.1–248.0)	244.5 (237.3–250.0)	245.8 (242.5–260.3)
	36	259.8 ^b^ (245.4–266.3)	249.3 (236.8–256.4)	253.5 (246.0–254.1)	254.8 ^b^ (251.3–271.5)
	43	269.3 ^b^ (254.8–275.1)	254.0 ^bc^ (244.3–265.0)	260.0 ^bc^ (254.6–265.0)	260.3 ^bc^ (257.0–277.0)
Food intake	15	13.9 (10.0–16.3)	14.7 (10.4–16.0)	15.6 (12.9–18.1)	14.0 (11.0–20.2)
	22	19.0 (17.1–21.5)	15.4 (13.6–18.7)	17.6 (14.3–21.6)	18.8 (16.8–21.2)
	29	19.3 (18.0–20.6)	18.5 (18.0–19.4)	18.6 (16.4–19.4)	20.1 (16.9–22.6)
	36	20.0 (17.1–21.3)	17.5 (17.3–18.1)	19.0 (17.4–19.5)	18.4 (16.3–19.9)
	43	16.6 (13.9–19.1)	15.6 (13.4–19.4)	17.3 (16.1–18.9)	18.4 (17.2–19.1)

Values are expressed as median with interquartile ranges. Group S, immobilization and spontaneous recovery group (*n* = 6); Group U, immobilization and therapeutic ultrasound group (*n* = 6); Group N, immobilization and NMES group (*n* = 6); Group UN, immobilization and therapeutic ultrasound and NMES combination group (*n* = 6). ^a^, Significant difference from Day 15 in the same group (*p* < 0.05); ^b^, Significant difference from Day 22 in the same group (*p* < 0.05). ^c^, Significant difference from Day 29 in the same group (*p* < 0.05).

**Table 4 vetsci-11-00158-t004:** Creatinine phosphokinase levels.

Parameter	Group			
	S	U	N	UN
Creatinine phosphokinase level (U/L)	420 (268–964)	534 (207–1061)	300 (228–554)	477 (297–852)

Values are expressed as median with interquartile ranges. Group S, immobilization and spontaneous recovery group (*n* = 6); Group U, immobilization and therapeutic ultrasound group (*n* = 6); Group N, immobilization and NMES group (*n* = 6); Group UN, immobilization and therapeutic ultrasound and NMES combination group (*n* = 6). The creatinine phosphokinase level was evaluated on Day 43.

**Table 5 vetsci-11-00158-t005:** Quadriceps muscle weight and quadriceps muscle measurements.

Quadriceps Muscle	Group					
	C	I	S	U	N	UN
Muscle weight/Body weight (mg/g)	7.9 (7.8–8.0)	6.1 (5.7–6.4)	8.4 ^#^ (8.0–8.9)	8.6 ^#^ (8.4–8.8)	9.0 *^#^ (8.8–9.0)	9.1 *^#^ (9.0–9.2)
Muscle measurement (mm)						
Length	29.3 (27.9–30.0)	23.1 * (22.1–23.8)	24.6 * (24.3–25.1)	25.5* (25.3–26.1)	27.5 ^#a^ (27.0–27.9)	28.6 ^#ab^ (27.7–28.9)
Width	14.2 (13.3–14.6)	12.0 * (10.8–12.6)	14.5 ^#^ (13.8–14.7)	13.8 ^#^ (13.0–14.7)	14.2 ^#^ (13.5–14.9)	14.5 ^#^ (13.5–14.9)
Height	11.9 (10.4–13.2)	8.7 * (8.1–10.0)	10.4 (9.4–11.0)	10.5 (10.2–11.1)	10.9 (10.7–11.4)	11.7 ^#^ (10.9–12.0)

Values are expressed as median with interquartile ranges. Group C, control group (*n* = 12); Group I, immobilization-alone group (*n* = 12); Group S, immobilization and spontaneous recovery group (*n* = 12); Group U, immobilization and therapeutic ultrasound group (*n* = 12); Group N, immobilization and NMES stimulation group (*n* = 12); Group UN, immobilization and therapeutic ultrasound and NMES combination group (*n* = 12). *, significant difference from Group C (*p* < 0.05); ^#^, significant difference from Group I (*p* < 0.05); ^a^, significant difference from group S (*p* < 0.05); ^b^, significant difference from group U (*p* < 0.05).

**Table 6 vetsci-11-00158-t006:** Histopathological score of quadriceps muscles.

Histopathological Parameter	Group					
	C	I	S	U	N	UN
Inflammation	0.0 (0.0–0.3)	2.0 * (1.8–2.3)	2.0 * (1.8–2.0)	0.0 ^#^ (0.0–1.0)	0.0 ^#^ (0.0–1.0)	0.0 ^#^ (0.0–1.0)
Vasculitis	0.0 (0.0–0.3)	2.0 * (1.0–3.0)	2.0 (0.8–3.0)	0.0 ^#^ (0.0–0.3)	0.0 ^#^ (0.0–0.3)	0.0 ^#^ (0.0–0.3)
Fibrosis	0.0 (0.0–1.0)	3.0 * (2.0–3.0)	1.0 (0.8–2.0)	0.0 ^#^ (0.0–1.0)	0.0 ^#^ (0.0–1.0)	0.0 ^#^ (0.0–0.3)
Muscle degeneration	1.0 (0.0–1.0)	2.5 * (2.0–3.0)	2.0 (1.0–2.0)	1.0 ^#^ (0.0–1.3)	1.0 ^#^ (0.8–1.0)	1.0 ^#^ (0.0–1.0)
Total score	0.5 (0.0–1.3)	9.0 * (8.0–10.3)	7.0 (5.0–7.3)	1.0 ^#^ (0.8–3.3)	1.5 ^#^ (1.0–2.3)	1.0 ^#^ (0.8–2.3)

Values are expressed as median with interquartile ranges. Group C, control group (*n* = 6); Group I, immobilization-alone group (*n* = 6); Group S, immobilization and spontaneous recovery group (*n* = 6); Group U, immobilization and therapeutic ultrasound group (*n* = 6); Group N, immobilization and NMES group (*n* = 6); Group UN, immobilization and therapeutic ultrasound and NMES combination group (*n* = 6); *, significant difference from Group C (*p* < 0.05); ^#^, significant difference from Group I (*p* < 0.05).

## Data Availability

The data presented in this study are available in the article.

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
