# Peer review of "Effects of Neuromuscular Electrical Stimulation and Therapeutic Ultrasound on Quadriceps Contracture of Immobilized Rats"

_vetsci, 2024, doi:10.3390/vetsci11040158_

Round 1

Reviewer 1 Report

Comments and Suggestions for Authors

Introduction:

Line 46, 47 and 48: these lines must be further developed and need more scientific references;

Line 48: it's not just the stifle joint that suffers from altered range of motion. It needs to be explained and complete in the manuscript.

Line 49: "previous studies" and it just has reference to number 5;

Line 52 and 53: it should be in the alignment of the line 47 and 48;

Line 55: needs more recents references;

Line 58 and 59: it does not have references/bibliography;

Line 93: The objectiva/aim and hypothesis needs to be restructured . On the hypothesis do not justify the reason, it will be discussed.

Material and methods

- Line 121: In group S, do not have a 28 day rehabilitation treatment started;

-Figure 1: It has to referred on the text;

- How did you perform immobilization? Briefly describe it;

- Line 141: on Group S, again, rehabilitation treatment was not started;

- Line 145: Explain the euthanasia on this group;

- Put in the text a image of ESTIMUS;

- Line 172: TENS mode, what is the type of current? 

- Line 194: change the sentence to: "were euthanized on day 43";

Results:

- Place throughout the text which statistical test was applied;

- check whether tables and figures are included in the text;

Discussion:

- a question arises, in clinical practice, where we cannot anesthetize in every treatment, since at that time our muscles are relaxed and we can gain benefits... comment and add to the discussion;

Author Response

We greatly appreciate the time and effort the reviewers put into improving our manuscript.

Reviewer 2 Report

Comments and Suggestions for Authors

This is an interesting topic to discuss, the authors have made a great job. It is a well-written and clear manuscript, easy to read, so I must congratulate them. However, I have some concerns that I would like to argue.

In page 3, section 2.1, I can not understand the purpose of making an immobilization-alone group, since you also have made as a control for the several treatments,the group S. And, besides, these rats were euthanized os day 14. Could you explain me?

In page 4, line 172, could you explain me how could you place the animals in a stifle neutral position if rats had been induced a quadriceps contracture?

In line 173, why did you choose this frequency for a TENS mode? From my point of view, this is a frequency used for a muscle strenght therapy, as you had been said in the introduction section. However, use of TENS mode implies different frequencies and it is used for pain treatment. Furthermore, it is supposed that the use of 1 mA of intensity has been chosen in order to standardize in all patients, even so, how were you sure about the efficacy of this therapy, because it seems a too low intensity for me?.

And also, in line 174, how did you choose the treatment frequency? In my opinion, if this therapy is used as an analgesic tool, it would be fine, or maybe it could be used every single day. However, if the purpose is to improve muscle strenght, maybe that frequency (50hz) applied during 5 days a week could aggravate the contracture.

In page 7, in Statistical section, why did you choose the Mann-Whtiney test instead of the Wilcoxon test to compare between two time points?

As you conclude in page 16, line 530, in the trial you found that stifle joint ROM improved without treatment, how do you think is this possible? Did they improve only with a daily movements trying to walk?

Mainly for this last issue, in my opinion, it is impossible to conclude (in line 545) that this manuscript could contribute for the treatment in dogs, as this situation is not real. In canine species the quadriceps contracture is very hard to treat, even in the early stages of the pathology, and these types of therapies maybe diminshed the time but they must be included in a much more comprehensive rehabilitation program. Despite this, it is really a very good work to analyze this two therapies. For that reason, you should change the last sentence of Conclusion section.

Eventually, congratulations to the authors for his amazing experimental work.

Author Response

(The authors gave the same response as above.)
